# 'Ghost' fossils of early coccolithophores point to a Triassic diversification of marine calcifying organisms

Sam M. Slater ⬢ [1] ✉, Isaline Demangel[1,2] & Sylvain Richoz[2]

Over geologic time, biocalcification – the process by which marine organisms make calcium carbonate ($CaCO_3$) – has reshaped climates, ocean life, and seawater chemistry. In particular, the evolution of coccolithophores, the largest group of nannoplankton and today's most productive calcifiers, transformed ocean environments and the carbon cycle. Their origins, however, remain enigmatic. This is partly because studying coccolithophore fossils traditionally requires $CaCO_3$ preservation. Here, we bypass this limitation, searching for their 'ghost' fossils –imprints on organic matter. We present coccolithophores from ~241-million-year-old (Triassic) rocks, predating previous records by ~26 million years (myrs). The >100 ghost fossils, exceptionally preserved within zooplankton faeces, show that coccolithophores, nannoplankton, 'modern' eukaryotic phytoplankton, and planktonic biocalcification evolved earlier than previously thought. Coccolithophores now first appear alongside stony corals and other unrelated calcifiers, suggesting a diversification of a range of marine calcifying organisms following Earth's deadliest mass extinction, the end-Permian event. These findings indicate that coccolithophore diversity remained remarkably low for ~50 myrs, until after the end-Triassic mass extinction, showing that both Triassic-bookending extinctions were critical in their evolution. Our discoveries elucidate the evolutionary origins of coccolithophores, but also highlight the role mass extinctions have played in shaping life on Earth.

A massive array of marine organisms, including corals, molluscs and crustaceans, build calcium carbonate ($CaCO_3$) skeletons and shells through the process of biocalcification. Today, most marine biocalcification takes place in the open ocean water column (the pelagic realm) by calcifying plankton[1], however, this was not always the case. Throughout the Palaeozoic, ~539–252 million years (myrs) ago, nearly all $CaCO_3$ deposition occurred in shallow water environments[2,3]. It was not until the Triassic–Jurassic transition, ~201 myrs ago, that biocalcification became widespread in the pelagic realm, driven by the evolution of calcareous nannoplankton[3]. This development fundamentally changed seawater chemistry, long-term biogeochemical cycles and

contributed to the formation of modern marine ecosystems; it therefore represents one of the most important evolutionary innovations in Earth's history[3–5]. Among calcareous nannoplankton, coccolithophores —a group of pelagic single-celled photosynthetic eukaryotic algae—are overwhelmingly dominant[6], representing the most productive calcifiers in today's oceans[7]. They promote cloud formation[8], play an important role in the carbon cycle[7], and are a major contributor to marine primary production[7]. Traditionally, coccolithophores were thought to have evolved near to the end of the Triassic Period[9], after calcareous dinoflagellates (a different group of calcareous nannoplankton) and nannoliths (minute $CaCO_3$ remains of uncertain

[1]Department of Palaeobiology, Swedish Museum of Natural History, Stockholm, Sweden. [2]Department of Geology, University of Lund, Lund, Sweden.
✉e-mail: sam.slater@nrm.se

biological affinity)[4,10]. Indeed, to our knowledge, prior to this study, the earliest coccolithophore fossil was a ~215-myr-old indeterminate coccolith from the Norian Stage of the Late Triassic[9]. Given that coccolithophore fossils have been studied for almost 200 years[11], with their fossils perhaps being the most numerous and stratigraphically complete of any group[12], these views are widely accepted.

Until now, the early fossil record of coccolithophores has exclusively relied on the preservation of 'body' fossils—the CaCO$_3$ remains of their 'exoskeletal' plates, called coccoliths[6] (Fig. 1a, b). Unfortunately, even when conditions favour the deposition and burial of CaCO$_3$ at the seafloor, CaCO$_3$ can dissolve later due to heat, pressure, and acidic pore-water infiltration into rock strata[13]. Given that older rocks have experienced more time for CaCO$_3$ dissolution to take effect, the preservation potential of CaCO$_3$, and therefore body fossils, generally diminishes with age. Hence, older rocks that hold the answers to understanding the origins of coccolithophores are poorly positioned to do so from a taphonomic perspective. Here, we applied a methodological approach never before conducted on Triassic rocks, bypassing the need for CaCO$_3$ body fossils. We primarily looked for their 'ghost', or imprint, fossils[13,14] preserved on organic matter (Fig. 1c). We sampled Middle (Ladinian) and Upper (Norian) Triassic rocks from Switzerland and Austria, and where possible, selected sedimentary rocks interbedded with volcanic ash layers that have been radiometrically dated and thus have precise age estimates[15] (Fig. 2, Supplementary Fig. 1 and Supplementary Data 1). After extracting

organic matter from rocks, we searched for ghost fossils using a scanning electron microscope (see 'Methods' for details).

## Results and discussion

### Ghost and body fossils

Ghost fossils of coccoliths were present in five samples (1, 2, 5, 6, 10) (Figs. 2 and 3 and Supplementary Figs. 1–3). To our knowledge, these are now the earliest known coccoliths in the fossil record, with the oldest predating the previous earliest coccolith by ~26 myrs[9]; they represent the first record of coccoliths from the Ladinian Stage and the Middle Triassic, and the first ghost fossils from the Triassic. Our discovery of Ladinian coccoliths also implies that coccolithophores must have existed throughout the Carnian, despite no previous reports of coccoliths from that stage. The ghost fossils are small, with coccoliths averaging 2.55 μm in length (n = 50). This partly explains their lack of discovery until now, but also reveals that in their early history, coccoliths started small, increasing in size with time.

All identifiable ghost fossils belonged to the coccolith species, *Crucirhabdus primulus* (e.g. Fig. 3b), except for one specimen of *?Archaeozygodiscus koessenensis* (Fig. 3f). The oldest sample (1), collected 23 cm below a volcanic ash bed 241.07 ± 0.13 myrs old (Ladinian)[15], yielded indeterminate coccoliths (e.g. Fig. 3a) and *C. primulus* (Supplementary Fig. 2), which to our knowledge now represent the earliest records of this species by ~32 myrs[9]. Samples 2 and 5 only yielded indeterminate coccoliths (Supplementary Fig. 2), but sample 6,

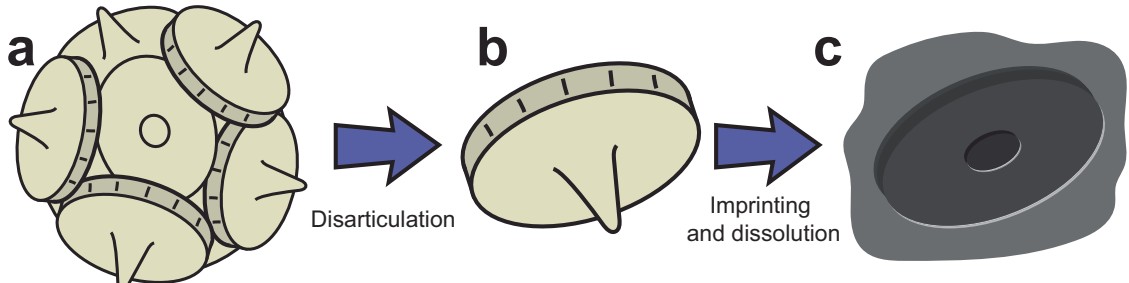

**Fig. 1 | Schematic of different types of coccolithophore fossils. a** Complete coccolithophore exoskeleton, made of coccoliths (individual plates); composed of calcite. **b** Coccolith 'body' fossil; composed of calcite. **c** Coccolith 'ghost' fossil;

imprint of coccolith on organic matter after calcite dissolution (see Fig. 3 for photographed examples); note the hole in the centre of the imprint formed from the spine of the coccolith.

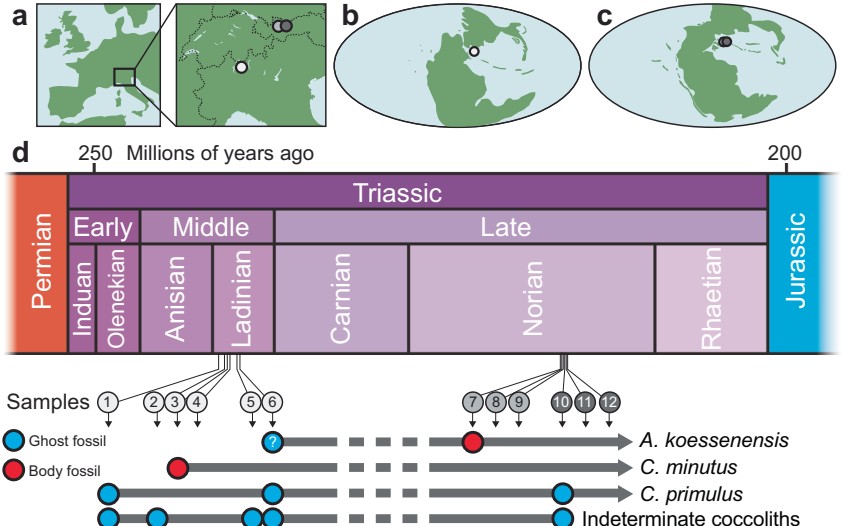

**Fig. 2 | Study sites and fossil occurrences. a** Study site locations. **b** Global palaeogeographic map showing the location of the Ladinian study site. **c** Global palaeogeographic map showing the location of Norian study sites. **d** Age of samples

and fossil occurrences. Palaeogeographic maps modified from ref. 47. For further sampling details, see 'Methods', Supplementary Fig. 1 and Supplementary Data 1.

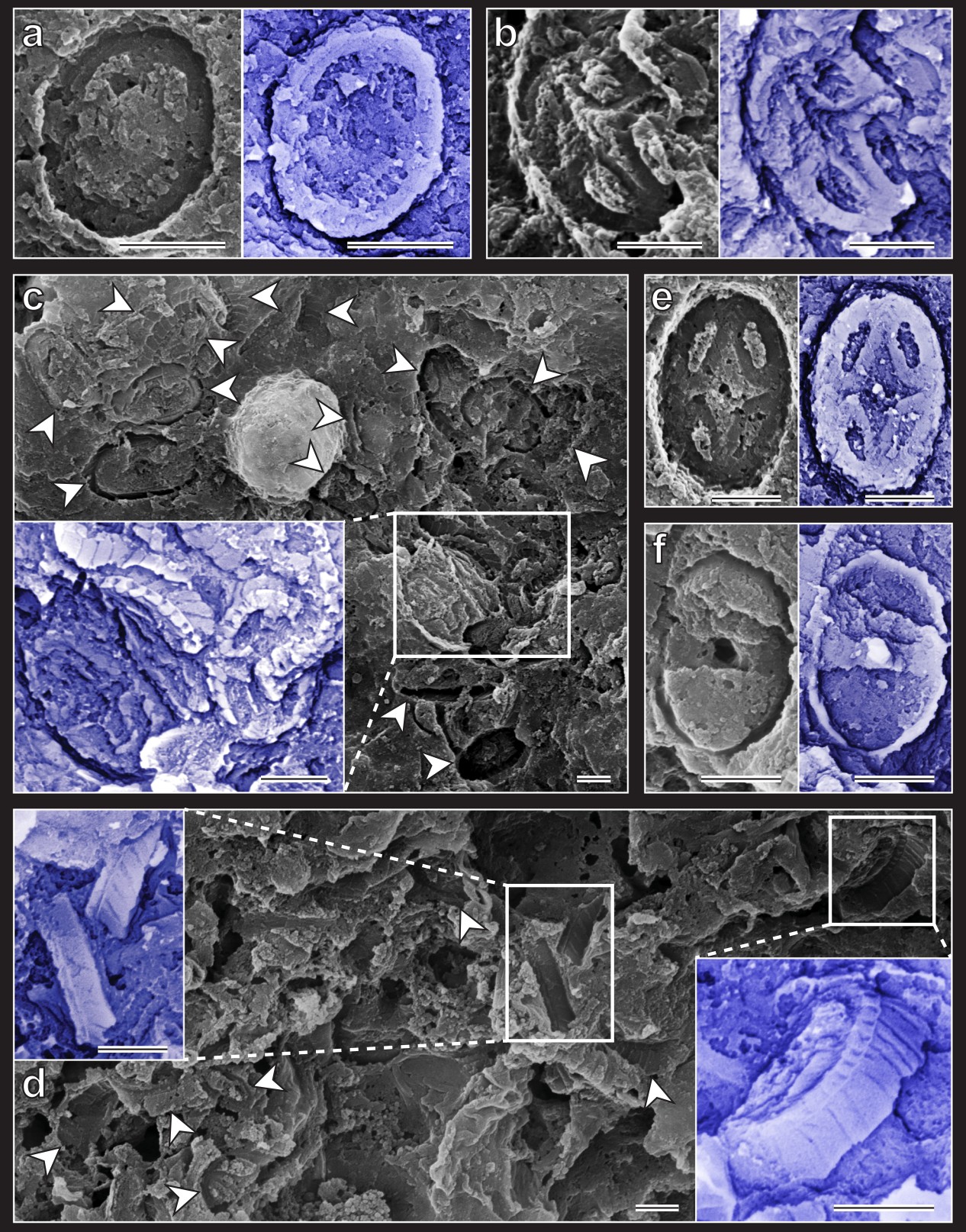

**Fig. 3 | Selected scanning electron microscope images of coccolithophore 'ghost' fossils. a** Indeterminate coccolith, sample 1 (Cava superiore 1; S206000-03); along with other coccoliths from this sample, to our knowledge this is now the earliest coccolith in the fossil record. **b** *Crucirhabdus primulus*, sample 6 (Val Mara D1; S206003-03). **c, d** Several *C. primulus* (arrows), sample 6 (Val Mara D1; S206003-03). **e** *C. primulus*, sample 6 (Val_Mara_D1ii; S206003-05).

**f** *?Archaeozygodiscus koessenensis*, sample 6 (Val_Mara_D1ii; S206003-05). Each sample was surveyed for 4 h. Blue images are inverted to better visualise the original coccoliths. Note that, compared to body fossils, imprinted and inverted images need to be interpreted with additional care, especially regarding features such as clockwise versus anticlockwise imbrication direction. Scale bars, 1 μm.

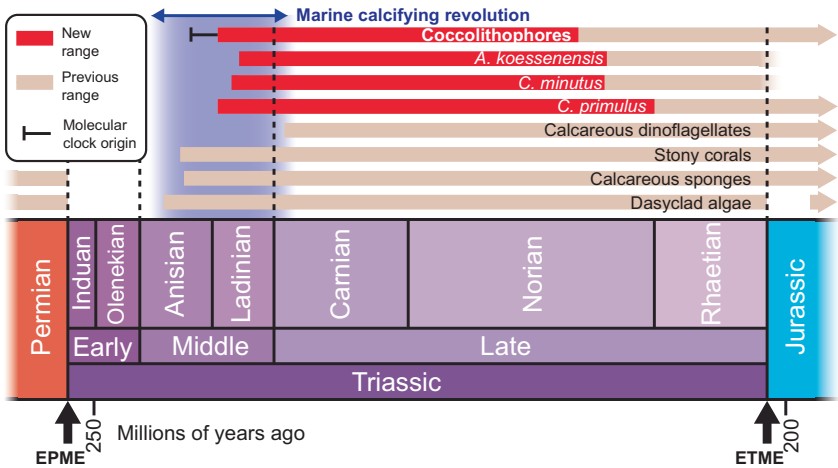

**Fig. 4 | Timeline of selected marine calcifying organisms.** EPME end-Permian mass extinction, ETME end-Triassic mass extinction. See Supplementary Data 2 for precise range details.

taken 10 m below a volcanic ash layer 239.51 ± 0.15 myrs old[15] (Ladinian; Supplementary Fig. 1), was the most productive sample. This yielded >70 *C. primulus* specimens, one *?A. koessenensis* and several indeterminate coccoliths (Fig. 3b–f, Supplementary Figs. 2 and 3). The *?A. koessenensis* is now the earliest specimen of this species to our knowledge, predating previous records by ~27 myrs. Preservation in sample 6 was exquisite, with imprinted coccolith spines, central area structures and rim elements (Fig. 3). Sample 10, the youngest productive ghost fossil sample, yielded two *C. primulus*, and several indeterminate, specimens (Supplementary Fig. 3); although these are the youngest *C. primulus* here, they are still ~7 myrs older than the previous earliest records of this species[9].

We also searched for body fossils from the same, plus several more CaCO₃-rich rock samples (Supplementary Fig. 1), finding one *Crucirhabdus minutus* (not found as a ghost fossil) in sample 3, and one *A. koessenensis* in sample 7 (Supplementary Fig. 4). To our knowledge, the *A. koessenensis* is now the earliest body fossil of its species and the *C. minutus* is the earliest body or ghost fossil of its species, predating previous records by ~27 myrs (Supplementary Data 2). The *C. minutus* specimen is now to our knowledge also the earliest coccolith body fossil in the entire fossil record, by ~25 myrs.

Taken alone, our body fossil finds would have substantially pushed back the initial occurrence of coccolithophores, however, our oldest ghost fossils push back these records further still. Furthermore, only two body fossils were found compared to >100 ghost fossils, which include numerous well preserved specimens (Fig. 3). No samples yielded both ghost and body fossils, and of the three species documented here, only *A. koessenensis* (one *?A. koessenensis* ghost fossil and one *A. koessenensis* body fossil) was found in both records (Fig. 3f and Supplementary Fig. 4). Our results clearly demonstrate the importance of ghost fossil analysis, but the differences between ghost and body fossil records show that each record is selectively capturing certain species, highlighting the utility of a combined approach.

## Zooplankton predation

The ghost fossils were all found on a type of 'smooth' amorphous organic matter (AOM), compared to a more abundant 'angular' type (Supplementary Fig. 5 and Supplementary Movie 1). Although the origins of the angular AOM are unclear, the smooth fragments resemble zooplankton faecal pellets[16]. Zooplankton ingestion increases the likelihood of coccolith fossilisation, as faecal pellets provide protection and additional ballast, enabling coccoliths to sink to the seafloor[6,17]. Clustered and overlapping imprints (Fig. 3c, d) represent recently (at the time of fossilisation) collapsed complete coccolithophore exoskeletons.

This suggests that the coccolithophores were eaten whilst alive; if they had died before ingestion, their articulated exoskeletons would have likely dispersed, breaking apart in the water column and degrading, or preserving as isolated imprints. Faecal pellets containing coccoliths also demonstrate that, by the Middle Triassic, coccolithophores were acting as primary producers at the base of marine food chains, transferring energy to higher trophic levels.

## *Crucirhabdus*—the common ancestor of coccoliths

Our results reveal a hidden early fossil record of coccoliths, demonstrating that the three most ancient species—*C. primulus*, *C. minutus* and *A. koessenensis*—have substantially longer ranges than previously reported (Fig. 4). In particular, *C. primulus* now has an exceptionally long range of >58 myrs[13]. Before this study, the initial occurrences of *C. minutus* and *A. koessenensis* were in the late Norian, and *C. primulus* was in the early Rhaetian[9]. Based partly on these initial occurrences, *C. minutus* was thought to be ancestral to *C. primulus*[9,18–21]. Our records reveal a different initial occurrence order of: (1) *C. primulus*, (2) *C. minutus*, and (3) *A. koessenensis*. At face value, this suggests that *C. primulus* was ancestral to *C. minutus*, but we emphasise that comparing these initial occurrences is imperfect, as *C. primulus* is restricted to ghost, and *C. minutus* is restricted to body, fossil records here. However, since the genus *Crucirhabdus* now predates all other coccolith genera in both ghost and body fossil records, our results suggest that this genus is ancestral to all known coccoliths.

## Phytoplankton evolution and marine calcifying revolution

Among the eukaryotic phytoplankton that produce readily fossilisable remains, today, three groups—dinoflagellates, coccolithophores and diatoms—known as the 'modern' eukaryotic phytoplankton, carry out the vast majority of the export flux of organic matter to the ocean interior and higher trophic levels[22]. Traditionally, dinoflagellates were thought to be the first of these groups to evolve[22]. However, our records challenge this view, showing that coccolith ghost fossils, to our knowledge, are now the earliest unequivocal fossils of modern eukaryotic phytoplankton, predating the earliest dinoflagellate fossils[23,24]. Our results therefore rewrite the appearance order of modern eukaryotic phytoplankton to: (1) coccolithophores, (2) dinoflagellates, and (3) diatoms. Our findings also overturn the paradigm that calcareous dinoflagellates and nannoliths[10,12] were the first calcifying plankton, which until this study, predated coccolithophores by ~21 myrs[9]. Instead, our results indicate that coccolithophores were the first pelagic plankton to develop biocalcification, a process that transformed marine environments and the carbon cycle.

Molecular clock predictions have placed the origin of coccolithophores between ~270–200 myrs ago[25,26], however, recalibration has refined this to ~243 myrs ago[27] (Fig. 4). Given that molecular clocks regularly predict origination dates significantly earlier than the earliest known fossils of many groups[28], it is notable to find fossils here that agree with, and even predate some, clock-based estimates. In this respect, coccolithophores are unusual, as unlike most present-day highly diverse clades[29], their crown group diversified remarkably slowly in their early history. Indeed, the scarcity of Triassic coccolith-derived limestones and chalks[30,31] indicates that, although present, coccolithophores remained a somewhat subsidiary component of marine ecosystems until after this period. It was not until the Jurassic, ~50 myrs after their inferred Early–Middle Triassic origin, that coccolithophores speciated more rapidly[32,33], suggesting that the end-Triassic mass extinction (ETME; ~201 myrs ago) was critical in giving rise to the requisite ecological and/or environmental changes that enabled their subsequent diversification.

The initial occurrence of coccolithophores is now ~10–11 myrs after the onset of Earth's deadliest event[34], the end-Permian mass extinction (EPME; ~252 myrs ago). Due to the severity of the EPME, marine ecosystems took ~5–10 myrs to recover[35–37], thus our coccolithophore initial occurrence coincides with the upper estimates of this recovery phase. Several other highly diverse extant clades first appear in the aftermath of mass extinctions, including famously, the placental mammals[38] and modern birds[39], which both appeared after the end-Cretaceous mass extinction (~66 myrs ago). Our ghost fossil record now demonstrates that coccolithophores also follow this pattern. Perhaps most significantly however, we note that their initial occurrence also coincides with the initial occurrences, and post-EPME Lazarus taxa reoccurrences, of several unrelated marine calcifiers, including the stony (scleractinian) corals[40], calcareous dinoflagellates[10], heavily calcified sponges, and calcifying green algae[41] (Fig. 4). Given that coccolithophores are today's most productive marine calcifiers[1], and stony corals are the most productive in coastal seas[42]—and incidentally, are the most important for maintaining marine biodiversity by constructing reefs[43]—these evolutionary developments would go on to overhaul ocean life, reshaping near- and offshore environments. Here, we propose this as the early Mesozoic marine calcifying revolution (Fig. 4). Since the various groups involved are phylogenetically unrelated but biochemically similar, we infer that extrinsic factors related to the reorganisation of marine ecosystems and changes to ocean chemistry after the EPME were central to their contemporaneous emergence. Specifically, the extinction of numerous Permian calcifiers[44] and the disappearance of complex reefs[45], which were the principal zones of marine biocalcification through the Palaeozoic[46], removed competitors and increased the availability of carbonate ions in Early–Middle Triassic oceans. Surviving lineages, including the soft (i.e. they lacked a hard $CaCO_3$ exoskeleton) precursors to coccolithophores, were then able to exploit this resource, giving rise to a roughly simultaneous diversification event across multiple calcifying lineages (Fig. 4).

Our findings show that coccolithophores originated much earlier than previously thought. Their post-EPME emergence and post-ETME diversification indicate that both Triassic-bookending extinctions were pivotal in their evolution, but also reveal how these now ubiquitous plankton persisted at low diversity for a substantial portion of their early history. Although it is surprising that these ancient coccoliths have eluded discovery until now, all prior attempts to find the earliest forms have only looked for their $CaCO_3$ body fossil remains. Our discoveries, which were made possible using the unconventional approach of ghost fossil analysis, demonstrate the potential of this method. More broadly, our study highlights that even intensively studied fossil groups, such as coccoliths, still have much to be discovered, but that innovative methodological approaches are needed to reveal such hidden records.

## Methods

### Study sites
We sampled three localities: (1) Monte San Giorgio, in the Southern Alps, Switzerland, and (2) Hahntennjoch and (3) Seefeld, in the Seefeld Basin, Northern Calcareous Alps, Austria. See Supplementary Fig. 1 and Supplementary Data 1 for sampling details. Samples in Austria were taken near roads in the public domain. No authorisation to sample and to export was necessary. The Monte San Giorgio area is a protected UNESCO site. Authorisation to sample and export was obtained from Dr. Rudolf Stockar, Cantonal Museum of Natural History, Lugano, Switzerland.

### Ghost fossil analysis
Eight rock samples of ~20 g were dissolved in 40% hydrochloric and 40% hydrofluoric acids to remove carbonates and silicates, respectively, in order to isolate organic matter. These methods are standard processing techniques in the field of palynology, i.e. to extract spores, pollen and marine organic-walled microfossils from sedimentary rocks. Residues were then sieved at 10 μm, and the resultant material was pipetted directly onto aluminium SEM stubs. Residues were left to dry for 24 h and gold-coated for 180 s. Examination of organic matter was conducted using an ESEM FEI Quanta FEG 650 SEM at the Swedish Museum of Natural History, Stockholm. Organic matter was observed in traverses, between ×5000 and ×10,000 magnifications to search for ghost fossils. Photographs were taken using higher magnifications. Each sample was surveyed for 4 h.

### Body fossil analysis
Twelve samples were prepared and examined for body fossils. Fresh rock surfaces of 1 square cm's were cut and polished with green silicon carbide powder at 800-mesh per inch using distillate water. The blocks were etched for 15 s in 0.1% hydrochloric acid and briefly cleaned in an ultrasonic bath with distilled water. Samples were then dried overnight at 50 °C and coated with a 1 nm (20 s) layer of platinum/palladium using Cressington Sputter Coater 208HR. SEM observations were conducted with a TESCAN MIRA 3 SEM at Lund University, Sweden, and each sample was examined for 4 h.

### Organic matter (palynofacies) analysis
To determine the predominant types of substrates onto which ghost fossils preserve, the proportions of different types of organic matter were counted using a light microscope (LM). Surplus organic matter residue from ghost fossil analysis was strewn across LM coverslips and mounted onto glass slides with epoxy resin. A minimum of 500 organic particles were counted per sample; see Supplementary Data 3 for organic matter categories; see Supplementary Note 1 for further details of organic matter (palynofacies) analysis.

### Reporting summary
Further information on research design is available in the Nature Portfolio Reporting Summary linked to this article.

## Data availability
All data generated or analysed during this study are included in this published article and its Supplementary Information files. All materials are housed in the collections of the Department of Palaeobiology at the Swedish Museum of Natural History, Stockholm (further details, including sample numbers are provided in Supplementary Data 1); contact the corresponding author for access.

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

## Acknowledgements
This research was funded by the Swedish Research Council (2019-04524 [S.M.S.], 2023-03330 [S.M.S.], 2022-04969 [S.R.]), Formas (Swedish Research Council for Sustainable Development; 2023-00984 [S.M.S.]), the Austrian National Committee for the International Geoscience Programme (Austrian Academy of Sciences [S.R.]) and the European Research Council under the European Union's Horizon 2020 Research and Innovation Program (833454 DEVOCEAN [Conley]). S.R. is grateful to R. Brandner, T.J. Algeo and R. Stockar for their help during sampling.

## Author contributions
S.M.S. and S.R. conceived the research. S.R. conducted the fieldwork. S.M.S. and I.D. performed the experiments and collected the data. All authors analysed the data and wrote the manuscript.

## Funding

## Competing interests
The authors declare no competing interests.
