## [Transparent Peer Review file · Nature Communications]

‘Ghost’ fossils of early coccolithophores point to a Triassic diversification of marine calcifying organisms

Corresponding Author: Dr Sam Slater

Version 0:

Reviewer comments:

Reviewer #1

(Remarks to the Author)

This is a first rate piece of work applying an interesting, but challenging, new technique, study of coccolith imprints in organic matter, to address a long-standing problem, determining the earliest evolutionary history of coccolithophores. This work was almost certainly laborious and time-consuming but they obtained some very useful new data. The paper is well illustrated and written, although it could be phrased more formally in places. I have made some annotations on the ms.

The drawing of a coccosphere (fig 1) is currently misleading since it is clearly based on *Crucirhabdus* but has the wrong type of rim. It should be amended to either have the correct rim, or to clearly be a fictional coccolith with no obvious similarity to any of the Triassic species.

One issue which is not mentioned in the paper is that the inverted images need to be interpreted with care as chiral features are reversed if the image is assumed to show positive relief. I noticed this because the imbrication direction shown in the rim of *C. primulus* appears to be the inverse of that normally seen (anti-clockwise instead of clockwise). This imbrication is more marked than is shown in most Jurassic specimens.

Jeremy Young

Reviewer #2

(Remarks to the Author)

The manuscript deals with a very interesting finding that regards the appearance of coccoliths in the stratigraphic record, namely the first occurrence of the Coccolithophores phytoplankton algae that are the most productive calcifiers in the ocean today as they have been so through the Mesozoic and Cenozoic. Previously, the earliest coccolithophore fossil was documented in a ~215-Myr-old sediment from the Late Triassic (Norian). In this manuscript, the Authors document the early record of Coccolithophores using “new and unconventional methods” for sample processing and analyses in older (accurately dated) rocks from Switzerland and Austria. They were able to find imprints of the earliest coccolithophores from ~241-Myr-old rocks (Middle Triassic, Ladinian), thus updating the evolutionary origins of coccolithophores (predating previous records by ~26 Myrs). This is a noteworthy result of this study.

Moreover, they highlight the role played by Triassic mass extinctions either in the origination of coccoliths and in the subsequent evolutionary development (i.e. the increase in diversity and abundance), the latter representing a fundamental step in the evolution of marine ecosystems and long-term biogeochemical cycles.

The paper is properly organized, it is concise and clearly written using correct English, references are up-to-date and relevant (except for a quoting in support of a statement, reported below and in the .docx file). The quality of illustrations is good (I only have a comment for Fig. 1c – see below).

A relevant aspect of this work is the methodological approach used for the performed analysis, that is a novelty for the micropaleontological study of Triassic rocks: through their search of “ghost” nanofossils, namely their imprints in organic matter, the Authors clearly widen the field of micropaleontologic analysis of nanofossils for potential future new discoveries. The interesting topic and the generally good quality of the manuscript make it suitable for the journal. It is in an almost proper shape for publication: I only have few notes and comments, reported below (and in the .docx file).

Line 54: Delete “intensively”. Even if the coccoliths have been discovered almost 200 years ago, they have been thoroughly

studied starting from the 50ties of the twentieth century.

Fig. 1: This figure is only partially meaningful: the representation of what is an imprint of coccolith, given in "1 c", is not at all explicative. I suggest to including one of the real images for showing what is a "ghost" coccolith (as one of those reported in Figure 3), because the black drawn coccolith specimen doesn't make sense.

Notes about chapter "Zooplankton predation"

Line 159 - The original paper in which it has been shown that coccoliths can be deposited on/in the seafloor through faecal pellets is by S. Honjo (1976) [Honjo, S., 1976. Coccoliths: production, transportation and sedimentation. *Marine Micropaleontology*, v. 1, 65-79].

Lines 164-171 - I found this part rather speculative: I do not think it is easy (reasonable) to identify what could be the "preferential food" taking into account only the very few species preserved (recognized); or to pretend to ascertain, from such a limited dataset of taxa, the variability of water depths inhabited by those "primordial" coccoliths. The presented data of presence or absence in the faecal pellets of the recognized species are limited and (as suggested by the Authors' assessment in lines 263-264) those coccoliths have still much to be discovered.

Reviewer #3

(Remarks to the Author)

The manuscript by Slater et al. demonstrates that Coccolithophores, important marine calcifying phytoplankton, originated much earlier than previously thought, in the Middle Triassic (~241 million years ago), not the Late Triassic as earlier believed.

This discovery was made using a novel method: instead of relying on fragile calcium carbonate (CaCO₃) body fossils, Slater et al. identified "ghost fossils", imprints of coccoliths preserved on amorphous organic matter. These ghost fossils are now the oldest known "coccoliths", predating earlier records by ~26 million years. These phytoplankton were found in zooplankton fecal pellets, indicating they were already part of marine food chains and primary producers in the Middle Triassic.

The newly identified early species include *Crucirhabdus primulus*, *C. minutus*, and *Archaeozygodiscus koessenensis* and suggest longer evolutionary ranges and a slower initial diversification of coccolithophores.

Data suggest that coccolithophores now represent the earliest known modern eukaryotic phytoplankton, predating dinoflagellates and diatoms. They were also the first pelagic calcifying plankton, overturning previous ideas that calcareous dinoflagellates came first. Their emergence after the end-Permian mass extinction and diversification after the end-Triassic mass extinction suggests that global ecological disruptions played a pivotal role in their evolution.

The manuscript is excellently written and very well illustrated. The interpretations are supported by the data. The work significantly expands our understanding of the evolution of marine primary producers and is therefore of great interest to geoscientists as well as scientists in other fields.

I have only two minor comment.

Lines 84-85: What kind of extraction procedure was applied. Please clarify. Please also clarify why the samples were extracted. Is it because bitumen will mask the coccolith imprints in the kerogen? Was the kerogen concentrate extracted? Please clarify and add information to the method section.

Line 195: what about nanno- and pico-plankton that produce no hard parts. Marine phytoplankton is not only composed on shell-producing algae groups. Actually, non-shell procuring small phytoplankton groups are the most abundant phytoplankton groups in the oceans (based on their abundance). Non-shell forming phytoplankton groups are frequently overlooked when marine primary producers are discussed.

Version 1:

Reviewer comments:

Reviewer #1

(Remarks to the Author)

This is a revised ms. The original version was a well-presented ms reporting significant data and I recommended it for publication subject to minor corrections. Those corrections have been carried out carefully, and I am now pleased to recommend it for publication

Reviewer #2

(Remarks to the Author)

All the (few) suggestions and advices by me (as Reviewer 2) and the other reviewers have been followed and the manuscript is ready to be published.

Reviewer #3

(Remarks to the Author)

No further modification needed for my side.

Point-by-point response to the reviewers' comments

Reproduced verbatim.

Responses provided in red.

REVIEWER COMMENTS

Reviewer #1 (Remarks to the Author):

This is a first rate piece of work applying an interesting, but challenging, new technique, study of coccolith imprints in organic matter, to address a long-standing problem, determining the earliest evolutionary history of coccolithophores. This work was almost certainly laborious and time-consuming but they obtained some very useful new data. The paper is well illustrated and written, although it could be phrased more formally in places. I have made some annotations on the ms.

All of the suggested changes in the reviewer's annotated PDF have been implemented.

The drawing of a coccosphere (fig 1) is currently misleading since it is clearly based on *Crucirhabdus* but has the wrong type of rim. It should be amended to either have the correct rim, or to clearly be a fictional coccolith with no obvious similarity to any of the Triassic species.

Good point. We have redrawn this figure in the suggested style, as a fictional coccolith, so as to not cause confusion.

One issue which is not mentioned in the paper is that the inverted images need to be interpreted with care as chiral features are reversed if the image is assumed to show positive relief. I noticed this because the imbrication direction shown in the rim of *C. primulus* appears to be the inverse of that normally seen (anti-clockwise instead of clockwise). This imbrication is more marked than is shown in most Jurassic specimens.

Good point. A note regarding this point has now been added to the caption of Figure 3.

Jeremy Young

Thank you for your considerate and helpful review.

Reviewer #2 (Remarks to the Author):

The manuscript deals with a very interesting finding that regards the appearance of coccoliths in the stratigraphic record, namely the first occurrence of the Coccolithophores phytoplankton algae that are the most productive calcifiers in the ocean today as they have been so through the Mesozoic and Cenozoic. Previously, the earliest coccolithophore fossil was documented in a ~215-myrs-old sediment from the Late Triassic (Norian). In this manuscript, the Authors document the early record of Coccolithophores using "new and unconventional methods" for sample processing and analyses in older (accurately dated) rocks from Switzerland and Austria. They were able to find imprints of the earliest coccolithophores from ~241-myrs-old rocks (Middle Triassic, Ladinian), thus updating the evolutionary origins of coccolithophores (predating previous records by ~26 myrs). This is a noteworthy result of this study.

Moreover, they highlight the role played by Triassic mass extinctions either in the origination of coccoliths and in the subsequent evolutionary development (i.e. the increase in diversity and abundance), the latter representing a fundamental step in the evolution of marine ecosystems and long-term biogeochemical cycles.

The paper is properly organized, it is concise and clearly written using correct English, references are up-to-date and relevant (except for a quoting in support of a statement, reported below and in the .docx file). The quality of illustrations is good (I only have a comment for Fig. 1c – see below). A relevant aspect of this work is the methodological approach used for the performed analysis, that

is a novelty for the micropaleontological study of Triassic rocks: through their search of “ghost” nannofossils, namely their imprints in organic matter, the Authors clearly widen the field of micropaleontologic analysis of nannofossils for potential future new discoveries. The interesting topic and the generally good quality of the manuscript make it suitable for the journal. It is in an almost proper shape for publication: I only have few notes and comments, reported below (and in the .docx file).

All suggestions in the attached .docx file have now been incorporated into the updated manuscript.

Line 54: Delete “intensively”. Even if the coccoliths have been discovered almost 200 years ago, they have been thoroughly studied starting from the 50ties of the twentieth century.

Agreed. This suggestion has been implemented.

Fig. 1: This figure is only partially meaningful: the representation of what is an imprint of coccolith, given in "1c", is not at all explicative. I suggest to including one of the real images for showing what is a “ghost” coccolith (as one of those reported in Figure 3), because the black drawn coccolith specimen doesn’t make sense.

Fair point. Following these, and Reviewer 1’s comments on this point, we have revised this figure. It is now a schematic of a fictional coccolith taxon (so as to not to be confused with any particular species); we have redrawn part C, so that the imprint is clearer. We have also added the word ‘imprinting’ to the figure, to make this point clearer. Since we have multiple images of imprints in Figure 3, we prefer to keep the schematic part and the real images separate.

Notes about chapter “Zooplankton predation”

Line 159 - The original paper in which it has been shown that coccoliths can be deposited on/in the seafloor through faecal pellets is by S. Honjo (1976) [Honjo, S., 1976. Coccoliths: production, transportation and sedimentation. Marine Micropaleontology, v. 1, 65-79].

Good point. Citation added.

Lines 164-171 - I found this part rather speculative: I do not think it is easy (reasonable) to identify what could be the “preferential food” taking into account only the very few species preserved (recognized); or to pretend to ascertain, from such a limited dataset of taxa, the variability of water depths inhabited by those “primordial” coccoliths. The presented data of presence or absence in the faecal pellets of the recognized species are limited and (as suggested by the Authors’ assessment in lines 263-264) those coccoliths have still much to be discovered.

Fair point. We have now removed this sentence, as we agree, it is a little speculative.

Thank you for your careful and helpful review.

Reviewer #3 (Remarks to the Author):

The manuscript by Slater et al. demonstrates that Coccolithophores, important marine calcifying phytoplankton, originated much earlier than previously thought, in the Middle Triassic (~241 million years ago), not the Late Triassic as earlier believed.

This discovery was made using a novel method: instead of relying on fragile calcium carbonate (CaCO₃) body fossils, Slater et al. identified "ghost fossils", imprints of coccoliths preserved on amorphous organic matter. These ghost fossils are now the oldest known “coccoliths”, predating earlier records by ~26 million years. These phytoplankton were found in zooplankton fecal pellets, indicating they were already part of marine food chains and primary producers in the Middle Triassic.

The newly identified early species include *Crucirhabdus primulus*, *C. minutus*, and *Archaeozygodiscus koessenensis* and suggest longer evolutionary ranges and a slower initial diversification of coccolithophores.

Data suggest that coccolithophores now represent the earliest known modern eukaryotic phytoplankton, predating dinoflagellates and diatoms. They were also the first pelagic calcifying

plankton, overturning previous ideas that calcareous dinoflagellates came first. Their emergence after the end-Permian mass extinction and diversification after the end-Triassic mass extinction suggests that global ecological disruptions played a pivotal role in their evolution.

The manuscript is excellently written and very well illustrated. The interpretations are supported by the data. The work significantly expands our understanding of the evolution of marine primary producers and is therefore of great interest to geoscientists as well as scientists in other fields.

I have only two minor comment.

Lines 84-85: What kind of extraction procedure was applied. Please clarify. Please also clarify why the samples were extracted. Is it because bitumen will mask the coccolith imprints in the kerogen? Was the kerogen concentrate extracted? Please clarify and add information to the method section.

This information is included in the Methods section (i.e. line ~430 onwards). We have added an in-text link to the Methods at lines ~84-85. To clarify these points we have also provided more details in the Methods section.

Line 195: what about nanno- and pico-plankton that produce no hard parts. Marine phytoplankton is not only composed on shell-producing algae groups. Actually, non-shell procuring small phytoplankton groups are the most abundant phytoplankton groups in the oceans (based on their abundance). Non-shell forming phytoplankton groups are frequently overlooked when marine primary producers are discussed.

Good point. This sentence has been modified so it is clear that we are now only referring to forms that readily preserve in the fossil record.

REVIEWERS' COMMENTS

Reviewer #1 (Remarks to the Author):

This is a revised ms. The original version was a well-presented ms reporting significant data and I recommended it for publication subject to minor corrections. Those corrections have been carried out carefully, and I am now pleased to recommend it for publication

Thank you for reviewing our manuscript.

Reviewer #2 (Remarks to the Author):

All the (few) suggestions and advices by me (as Reviewer 2) and the other reviewers have been followed and the manuscript is ready to be published.

Thank you for reviewing our manuscript.

Reviewer #3 (Remarks to the Author):

No further modification needed for my side.

Thank you for reviewing our manuscript.